Genome-wide identification of the OMT gene family in Cucumis melo L. and expression analysis under abiotic and biotic stress

Wang Shuoshuo 1
Wang Chuang 2
Lv Futang 1
Chu Pengfei 1
Jin Han jinhan@lcu.edu.cn 1
1 Liaocheng University , Liaocheng , China
2 Liaocheng Vocational & Technical College , Liaocheng , China
Shrestha Jiban
Electronic publication date: 2023 Dec 14
Publication date: 2023
Volume: 11
Electronic Location ID: e16483
Received 2023 May 5; Accepted 2023 Oct 27
Copyright: ©2023 Wang et al.
Copyright year: 2023
Copyright holder: Wang et al.
License: This is an open access article distributed under the terms of the Creative Commons Attribution License, which permits unrestricted use, distribution, reproduction and adaptation in any medium and for any purpose provided that it is properly attributed. For attribution, the original author(s), title, publication source (PeerJ) and either DOI or URL of the article must be cited.
License URL: https://creativecommons.org/licenses/by/4.0/

Keywords: Melon, CmOMT genes, Phylogenetic analysis, Expression analysis, Stress

Funding: Key Research and Development Program of Shandong 2019YQ034 Liaocheng University, China 318052244 318052290 31946221226 Key Research and Development Program of Liaocheng 2022YDNY11 This work was supported by the Key Research and Development Program of Shandong (grant number 2019YQ034), Liaocheng University, China (grant numbers 318052244, 318052290 and 31946221226), and the Key Research and Development Program of Liaocheng (grant number 2022YDNY11). The funders had no role in study design, data collection and analysis, decision to publish, or preparation of the manuscript.

==============================
Background

O-methyltransferase (OMT)-mediated O-methylation is a frequent modification that occurs during natural product biosynthesis, and it increases the diversity and stability of secondary metabolites. However, detailed genome-wide identification and expression analyses of OMT gene family members have not been performed in melons. In this study, we aimed to perform the genome-wide identification of OMT gene family members in melon to identify and clarify their actions during stress.

Methods

Genome-wide identification of OMT gene family members was performed using data from the melon genome database. The Cucumis melo OMT genes (CmOMTs) were then compared with the genes from two representative monocotyledons and three representative dicotyledons. The basic information, cis-regulatory elements in the promoter, predicted 3-D-structures, and GO enrichment results of the 21 CmOMTs were analyzed.

Results

In our study, 21 CmOMTs (named CmOMT1-21) were obtained by analyzing the melon genome. These genes were located on six chromosomes and divided into three groups composed of nine, six, and six CmOMTs based on phylogenetic analysis. Gene structure and motif descriptions were similar within the same classes. Each CmOMT gene contains at least one cis-acting element associated with hormone transport regulation. Analysis of cis-acting elements illustrated the potential role of CmOMTs in developmental regulation and adaptations to various abiotic and biotic stresses. The RNA-seq and quantitative real-time PCR (qRT-PCR) results indicated that NaCl stress significantly induced CmOMT6/9/14/18 and chilling and high temperature and humidity (HTH) stresses significantly upregulated CmOMT14/18. Furthermore, the expression pattern of CmOMT18 may be associated with Fusarium oxysporum f. sp. melonis race 1.2 (FOM1.2) and powdery mildew resistance. Our study tentatively explored the biological functions of CmOMT genes in various stress regulation pathways and provided a conceptual basis for further detailed studies of the molecular mechanisms.

Introduction

Melon (Cucumis melo L., 2n = 24) is an economically important and widely cultivated crop, and it ranks ninth in terms of global production among horticultural crops (Kong et al., 2016). It is a sweet, musky, fleshy fruit rich in vitamins, minerals, and health-promoting antioxidants. More than 32 million tons of melon were produced worldwide in 2017 (Zhao et al., 2019). However, various stressors, such as disease, pest, drought, and salt stress, have caused steep declines in melon yield and quality during growth (Wang et al., 2022). Therefore, melons have evolved complex defense mechanisms to manage such stress (Jander & Clay, 2011). In general, plants protect themselves from adverse stresses in numerous ways, including programmed cell death (PCD), antimicrobial substance secretion, and secondary metabolite and endogenous hormone production (Jing et al., 2017; Kud et al., 2019). Plant hormones, such as abscisic acid, salicylic acid, ethylene, and melatonin, play essential roles in response to abiotic and biotic stresses (Li & Li, 2019; Zeng et al., 2022). Nevertheless, plant genomes have evolved to adapt to environmental changes (Kaessmann, 2010). The main factors underlying gene family evolution are gene duplication and loss events, and most duplications are generated via whole-genome duplication (WGD) and small-scale duplication events (Panchy, Lehti-Shiu & Shiu, 2016; Elisabeth, Leng & Alexander, 2018). Under abiotic and biotic stress conditions, members of various gene families execute specific biological functions that interconnect to form networks that control plant resistance (Glover, Redestig & Dessimoz, 2016; Wang et al., 2022) and govern a wide range of plant biochemical pathways (Glover, Redestig & Dessimoz, 2016). Investigating the gene families implicated in resistance mechanisms is crucial for exploiting biotechnological tools to improve desirable agronomic traits, such as crop growth and productivity. Therefore, the current study was performed to identify and clarify the actions of gene families in melon.

O-methyltransferases (OMTs) constitute one of the three major plant methyltransferases, and they transfer the methyl group of S-adenosyl-L-methionine (SAM) to the hydroxyl group of several natural compounds, thus forming methyl ether derivatives (Roje, 2006; Struck et al., 2012). Plant OMT genes can be divided into two main classes based on their molecular weight and divalent ion dependence: caffeic acid OMT (COMT) and caffeoyl coenzyme a OMT (CCoAOMT) (Liu et al., 2015). CCoAOMT and COMT participate in lignin biosynthesis. Unlike CCoAOMT, which acts on caffeinated CoA and 5-hydroxyferulic acid CoA, COMT can act on a range of substrates, such as chalcones, caffeic acid, 5-hydroxyferulic acid, and 5-hydroxyferuloylester (Roje, 2006; Liu et al., 2015). OMT coordinates the methylation of some secondary metabolites. Based on sequence homology and substrate differentiation, the OMT gene family can be divided into three categories that mediate the methylation of flavonoids, caffeoyl-CoA, and small phenolic compounds (Noel et al., 2003). FAOMT and VvAOMT were verified to catalyze anthocyanin methylation in red grapevines and grapevines, respectively (Hugueney et al., 2009; Lücker, Martens & Lund, 2010). Zhang et al. (2021a); Zhang et al. (2021b) demonstrated that AOMTs are primarily involved in new functionalization or non-functionalization after tandem duplication events. Many functional natural compounds, such as flavonoids, alkaloids, plant antitoxins, and lignin precursors, are generated through enzyme-catalyzed modifications in response to biotic and abiotic stresses (Pichersky & Lewinsohn, 2011; Gana et al., 2013).

Because OMT genes have essential roles in plant secondary metabolism, their functions have been extensively studied in numerous plant species (Sun et al., 2019; Chang et al., 2021). Several studies have shown that ASMT acts as an end enzyme and limits plant melatonin biosynthesis (Zhang et al., 2014). In the last two steps of melatonin biosynthesis, ASMT (N-acetylserotonin O-methyltransferase) methylates NAS (N-acetylserotonin) into melatonin in the cytoplasm (Byeon et al., 2014). The ASMT1 gene was cloned from Oryza sativa in 2015 and named OsASMT1, and studies have shown that OsASMT2/3 is upregulated in healing tissues in OsASMT2/3-overexpression transgenic rice plants (Park, Byeon & Back, 2013). Currently, ASMT genes have been cloned from a variety of dicotyledonous plant species, such as Arabidopsis and Hypericum perforatum (Byeon et al., 2016; Zhou et al., 2021). Melatonin (n-acetyl-5-methoxytryptamine) is an essential bioactive molecule commonly found in animals and plants (Hardeland, Cardinali & Srinivasan, 2011; Arnao, 2014). Numerous studies have shown that melatonin plays an important regulatory role in plant growth and development and responses to biotic and abiotic stresses, such as saline-alkali stress, chilling stress, and powdery mildew infection (Sun et al., 2019; Chang et al., 2021; Zeng et al., 2022). Melatonin improves salinity tolerance in tomato through the systematic regulation of ionic, water, acid–base, and redox balance (Yan et al., 2019). At the initial stage of pathogen infection, exogenous melatonin can increase intracellular H2O2 content, and it acts upstream of salicylic acid (SA) and positively regulates its accumulation to govern plant disease resistance (Zeng et al., 2022). COMT, a multifunctional enzyme also reported to have ASMT activity, is a member of the OMT family that can methylate numerous secondary metabolites, such as phenylpropanoids, flavonoids, and alkaloids (Lam et al., 2007; Byeon et al., 2014). In addition to determining the roles of COMT genes in the model crops Oryza sativa, Arabidopsis, and Solanum lycopersicum, their functions in melatonin biosynthesis in other plants must be further characterized to understand their novel functions (Byeon et al., 2014; Byeon et al., 2015; Liu et al., 2019). Additionally, COMT genes with ASMT functions can be applied to improve crop yield and quality through genetic manipulation and rational breeding.

The OMT gene family has been reported to be directly or indirectly involved in abiotic and biotic stress tolerance, and the specific functions of some OMTs have been identified in a variety of crops. In Solanum lycopersicum, overexpression of SlCOMT1 significantly increases melatonin accumulation and improves salt and drought tolerance (Sun et al., 2019). In Oryza sativa, OsCOMT3 plays an important role in enhancing nematode resistance (Petitot et al., 2017). A total of 58 OMT genes were identified in Citrus, 27 of which were involved in the methylation of flavonoids and contributed to the functional study of CitOMT genes. Phylogenetic analysis of OMT genes from three representative plants revealed that these proteins were classified into the COMT and CCoAOMT subfamilies (Liu et al., 2015). In soybean, 55 COMT genes were divided into two groups, and GmCOMT genes were differentially expressed under various abiotic stresses, such as salt and drought stress (Zhang et al., 2021a; Zhang et al., 2021b). In watermelon, a total of 16 ClOMT genes were identified and classified into three groups, and ClCOMT1 was significantly upregulated under multiple stress conditions (cold, drought, and salt stress), whereas in transgenic Arabidopsis, ClCOMT1 overexpression enhanced abiotic stress resistance (Chang et al., 2021). However, few studies have focused on the involvement of this gene family and its members in abiotic and biotic stresses and their regulation in melons.

In this study, we aimed to perform the genome-wide identification of OMT gene family members in melon to identify and clarify their actions during stress using data from the melon genome database. The objectives of this study were to (i) identify and analyze CmOMT genes, (ii) analyze the expression profile of CmOMT genes in melon tissues, (iii) analyze the response of CmOMT genes to abiotic and biotic stresses, (iv) elucidate the evolution the OMT gene family, and (v) provide a basis for the screening of melon stress-related CmOMT candidate genes and their regulation. We identified the phylogenetic relationships of CmOMT genes with their homologs from other plants as well as their chromosome location, gene structure, and amino acid composition. Thereafter, we performed RNA-seq and quantitative real-time reverse transcription-polymerase chain reaction (qRT-PCR) assays of CmOMT genes to determine their tissue-specific expression and responses to abiotic and biotic stresses.

Materials & Methods

Plant and material

‘Yangjiaomi’ is widely cultivated in China and has become an important commercial cultivar because of its crunchy texture, juiciness, and sweet taste. ‘Yangjiaomi’ was growing in the plant culture room of Liaocheng University, Liaocheng, China. Melon seeds were germinated in an incubator at 28 °C and then sown in 50-hole cavity trays (grass: vermiculite: perlite = 1:1:1, v/v). After the seedlings had grown three leaves and heart-shaped leaves, the treatments were applied in a plant culture room (26 °C/20 °C, day/night) with a 16 h photoperiod. The same melon plants were selected for the abiotic stress treatments, which were set according to a previous study: NaCl stress (300 mM for 1, 3, 5 and 7 d), chilling stress (6 °C for 6 and 12 h), and high temperature and humidity (HTH) stress (day/night temperature at 45 °C/35 °C, soil moisture at 100%, and humidity at 90%) (Wang et al., 2016; Diao et al., 2020; Weng et al., 2022). The control and treated melons were collected to determine the relative expression of the CmOMT gene family. Each stress treatment group included 15 melon plants, and three biological replicates were performed for all treatment groups.

Identification of O-methyltransferase (OMT) genes in Cucumis melo L.

The melon (DHL92) protein database (Melon protein v4.0) was downloaded from the Cucurbitaceae Genome Database (CuGenDB; http://cucurbitgenomics.org/). To comprehensively identify the melon OMT gene family members, we searched the NCBI (Conserved Domain Search, http://www.ncbi.nlm.nih.gov/Structure/cdd/wrpsb.cgi/) and PFAM websites (http://pfam.xfam.org/) and analyzed the conserved motifs and protein functions of the reported ASMT and COMT proteins. PF00891 was employed as the reference sequence for the Hidden Markov Model (HMM) analysis of the melon protein database, and proteins with an e-value ≤ 1 × e −10 were selected as candidate OMT proteins (Liu et al., 2015; Chang et al., 2021). The candidate proteins were subjected to protein sequence identification using the SMART website (http://smart.embl-heidelberg.de/) (Letunic, Khedkar & Bork, 2017). Sequences without OMT structural domains or incomplete structural domains were removed, and the longest sequence was retained if redundant sequences were identified.

Physicochemical properties and chromosomal mapping of CmOMT genes in Cucumis melo L.

The molecular weight, isoelectric point (PI), formula, instability index, aliphatic index, and grand average hydropathicity (GRAVY) of the CmOMT genes were analyzed using the online tool ExPASy (http://web.expasy.org/protparam/) with the default parameters. The melon genome website (http://cucurbitgenomics.org/organism/18) provides the position information of CmOMT genes using MapChart software to map the distribution of CmOMT genes on chromosomes. The identification criteria for tandem duplication genes were as follows: (i) adjacent homologous genes were located on the same chromosome, with only one intercalated gene in the middle, and (ii) the comparison length and similarity of the two gene sequences were more than 70% (Zhu et al., 2014; Zhao et al., 2018).

Phylogenetic analysis, exon-intron structure, and motif analysis

Multiple alignments of the amino acid sequences of the 21 CmOMT proteins were performed using the ClustalW tool with default parameters (Larkin et al., 2007). To investigate the phylogenetic relationships of OMT homolog protein sequences among six species (Arabidopsis thaliana, Cucumis melo L., Cucumis sativus L., Solanum lycopersicum L., Oryza sativa L., and Citrullus lanatus (Thunb.) Matsum. et Nakai), a phylogenetic tree was constructed using the neighbor-joining algorithm of MEGA X software. A phylogenetic tree was constructed using the neighbor-joining Jones Taylor–Thornton (JTT) matrix model in MEGA X software, where bootstrap repetitions were set to 1,000 to assess the plausibility of the evolutionary tree (Kumar, Stecher & Tamura, 2016). The phylogenetic tree was visualized using Figtree v1.4.3 (http://tree.bio.ed.ac.uk/software/figtree/).

The CmOMT gene structure was analyzed using the Gene Structure Display Server (GSDS) website (Hu et al., 2015) (http://gsds.cbi.pku.edu.cn). General Feature Format (GFF) annotations for melons were downloaded from the genome database. The required annotation content for the CmOMTs exon-intron structure was uploaded to the GSDS website. The online tool Multiple Em for Motif Elicitation (MEME: https://meme.nbcr.net/) was used to further identify and analyze the conserved protein motifs of the CmOMT gene family members, and a motif number of 5 was implemented (Bailey et al., 2009).

Promoter analysis, transmembrane helix prediction, and structure prediction for CmOMT proteins

The promoter sequences of the CmOMT genes were analyzed using PlantCARE (http://bioinformatics.psb.ugent.be/webtools/plantcare/html/). Briefly, we used the sequence 2,000 bp upstream of the ATG start codon of the CmOMT gene family as the promoter sequence. Cis-regulatory elements in CmOMT gene promoters identified using PlantCARE were conserved for visualization using the GSDS website. In standard mode, the three-dimensional (3-D) structure of each CmOMT protein was modeled using the PHYRE2 tool (http://www.sbg.bio.ic.ac.uk/phyre2/html/page.cgi?id=index).

GO term and expression analysis for CmOMT genes

Gene Ontology (GO) term analyses were performed for CmOMT genes using the R package (ClusterProfiler package, enrichplot package), and terms with Q-values ≤ 0.05 were considered significantly enriched.

The expression of CmOMT genes in various organs was examined using a previously published RNA-seq dataset (Yano et al., 2020). A heatmap was generated to illustrate the spatiotemporal expression in the callus, dry seeds, root, stem (downside and upside), shoot apex, leaves (young and 6th–12th), tendril, flower (anther male, petal female, and stigma female), ovary [0–4 DAF (days after flowering)], fruit flesh (8–50 DAF), and fruit epicarp (8–50 DAF). Published RNA-seq datasets analyzed by Sebastiani et al. (2017) and Zhu et al. (2018) were analyzed for Fusarium oxysporum f. sp. melonis race 1.2 (FOM 1.2) and powdery mildew, respectively. Published RNA-seq data were used to generate heatmaps and volcano plots using the R package (Heatmaply package) and online tool Dynamic volcanogram (https://www.omicshare.com/tools/), respectively. To verify the expression of CmOMT genes under different stress conditions, total RNA was extracted using the RNA isolator Total Extraction Reagent (Vazyme, Nanjing, China). The extracted RNA was reverse-transcribed to cDNA using the HiScript II First Strand cDNA Synthesis Kit (Vazyme, Nanjing, China). qRT-PCR assays were performed on a qTOWER3G Real-time System (Analytik Jena AG, Jena, Germany). The relative expression based on three biological and technical repeats was calculated by the 2−ΔΔCT method (Livak & Schmittgen, 2002). All primers used for qRT-PCR are listed in Table S1.

Statistical analysis

All data were analyzed using SPSS software version 26.0 (SPSS Inc., Chicago, IL, USA) and presented as the mean ± standard deviation (SD) of three biological repeats. Differences among the results were tested by one-way ANOVA (one-way analysis of variance), with a P-value ≤ 0.05 indicating significance.

Results

Identification and characterization of CmOMT genes in Cucumis melo L.

Twenty-one CmOMT candidate genes were identified in the melon genome (Table S2) and named CmOMT1 to CmOMT21 based on the chromosomal gene order. Certain CmOMT genes were also distributed in adjacent regions on the same chromosome (Table 1 and Fig. 1A). The genes presented an uneven distribution on 12 chromosomes and were present on six chromosomes; however, four CmOMTs were not mapped to any of the chromosomes (Fig. 1A). CmOMTs were intensively distributed on chromosome 1, three CmOMTs were distributed on chromosome 6, two CmOMTs were distributed on chromosomes 4 and 11, and only one CmOMT gene was distributed on chromosomes 7 and 10 (Fig. 1A). According to a previously defined tandem arrangement, CmOMT1–CmOMT4, CmOMT13, CmOMT14, and CmOMT18–CmOMT21 were not tandem duplicated genes, whereas other CmOMTs appear to be generated by tandem repeat events (Zhu et al., 2014; Zhao et al., 2018). For example, CmOMT5–CmOMT8 may belong to a tandem duplication gene cluster while CmOMT9–CmOMT12 may be part of another tandem duplication gene cluster.

Table 1 The basic information of CmOMT genes in melon.

OMT Member	gene ID	Number of amino acids	Molecular
weight	PI	Formula	Instability
index	Aliphatic
index	Gravy	
CmOMT1	MELO3C027330	251	27,891.46	9.13	C1261H1993N337O352S12	30.79	89.72	−0.139	
CmOMT2	MELO3C027370	145	16,247.76	5.04	C717H1147N193O216S10	36.73	92.83	−0.094	
CmOMT3	MELO3C000487	181	19,483.45	4.60	C882H1379N211O266S9	45.16	88.84	0.109	
CmOMT4	MELO3C028075	93	9,555.92	4.47	C418H663N107O134S7	31.49	79.57	0.280	
CmOMT5	MELO3C018855	238	26,895.11	4.76	C1225H1892N296O355S14	38.98	86.81	−0.092	
CmOMT6	MELO3C018856	238	26,659.81	5.04	C1207H1883N301O352S13	37.07	89.71	−0.061	
CmOMT7	MELO3C018858	261	28,713.79	4.59	C1293H1999N321O391S13	42.33	88.12	−0.002	
CmOMT8	MELO3C018859	384	43,523.50	5.43	C1966H3059N501O561S26	43.62	90.89	−0.079	
CmOMT9	MELO3C013310	358	40,115.49	5.70	C1798H2831N473O519S23	42.13	97.82	0.012	
CmOMT10	MELO3C013311	360	39,968.08	5.47	C1775H2802N466O531S25	42.99	90.97	−0.061	
CmOMT11	MELO3C013313	358	39,806.86	5.60	C1769H2807N481O522S20	41.38	98.32	−0.002	
CmOMT12	MELO3C013315	290	32,129.94	5.25	C1447H2275N373O428S12	31.14	97.83	−0.002	
CmOMT13	MELO3C026750	318	35,978.56	5.52	C1608H2527N423O470S21	58.95	92.58	−0.172	
CmOMT14	MELO3C009403	371	40,926.88	5.84	C1814H2860N490O545S21	41.61	84.93	−0.161	
CmOMT15	MELO3C014089	279	30,768.32	5.36	C1379H2167N351O416S14	36.72	92.62	0.002	
CmOMT16	MELO3C014091	359	39,670.30	5.14	C1780H2775N463O533S15	33.62	92.84	−0.030	
CmOMT17	MELO3C014098	349	38,759.51	5.41	C1742H2709N457O508S18	20.07	92.15	−0.004	
CmOMT18	MELO3C024861	359	39,323.66	5.89	C1774H2778N458O506S22	30.82	96.43	0.091	
CmOMT19	MELO3C018336	365	41,114.67	5.84	C1859H2909N483O528S20	43.08	95.07	−0.089	
CmOMT20	MELO3C019324	357	40,254.50	5.90	C1801H2852N472O531S20	45.21	93.42	−0.148	
CmOMT21	MELO3C013600	357	39,648.44	5.45	C1793H2769N461O522S16	42.55	92.66	−0.045	

Figure 1 Chromosome distribution of CmOMTs (A) and phylogenetic tree, gene structure and conserved motif distribution of CmOMT gene family members (B).

The chromosome location, strand, description, number of amino acids, molecular weight, PI, formula, instability index, aliphatic index, and grand average hydropathicity (GRAVY) characteristics of the CmOMT genes were obtained. The average amino acid sequence of the 21 CmOMT proteins was 298 amino acids, with the value ranging from 93 (CmOMT4) to 384 amino acids (CmOMT8) (Table 1). The average molecular weight of the 21 CmOMT proteins was 33,211.24, the PI was approximately 5.50, the instability index was approximately 38.88, and the aliphatic index was approximately 91.63 (Table 1). In addition, five of the CmOMT proteins (CmOMT3, CmOMT4, CmOMT9, CmOMT15, and CmOMT18) were hydrophobic (Table 1).

Analysis of the gene structure and motifs in CmOMT genes

To further investigate the CmOMT gene family, the UTR and CDS distribution and number of CmOMT genes were analyzed based on phylogenetic trees. On the same branch, the CmOMTs were divided into class I and class II, which contained nine and six CmOMTs, respectively (Fig. 1B). On the other branch, six CmOMTs were grouped into class III, which was distinct from classes I and II (Fig. 1B). Genetic structural characterization showed that the number of CDS in classes I and II was similar, with 1–3 CDS regions (Fig. 1B). Class III differed from classes I and II in terms of gene structure and included 4 CDS regions, excluding CmOMT18 (Fig. 1B). Each exon in the same sister gene pair was very similar and presented the same size. However, introns were more variable in sequence length than the strongly conserved exons of sister gene pairs. This result indicates that introns are evolutionarily more unstable than exons, which is consistent with traditional evolutionary theory (Zhang et al., 2021a; Zhang et al., 2021b).

Conserved protein motifs were identified in CmOMTs. Five different motifs with 15–34 amino acids were found in the CmOMT gene family (details on the conserved structural domains are illustrated in Fig. S1A). CmOMT protein sequences were compared using a phylogenetic tree, and the locations were mapped using clustering and motif analyses. Details of the CmOMTs are presented in Fig. S1B. Motif 1 was present in most of the CmOMTs except for CmOMT2 (Fig. 1B). Motif 2 was not present in CmOMT3, CmOMT4, CmOMT7, CmOMT12, and CmOMT15 (Fig. 1B). Motifs 3 and 4 represented the basis of the CmOMT structural domain because all CmOMTs contained motifs 3 and 4. Motif 5 was identified in only nine CmOMT members, which belonged to class I (CmOMT7, CmOMT8, CmOMT13, and CmOMT19) and class II (CmOMT9–CmOMT12 and CmOMT21) (Fig. 1B). These results suggested that there were significant differences between CmOMT proteins after whole genome duplication (WGD) events and that the loss and retention of motifs may be related to the evolution of subbranches.

Phylogenetic tree and structure analysis of OMT proteins

To investigate the evolutionary relationships of plant OMT homologs, 120 OMTs were identified in melon, Arabidopsis (Chen et al., 2021), cucumbers (Table S3), tomatoes (Lu et al., 2019), rice (Liang et al., 2022) and watermelon (Chang et al., 2021), and an unrooted phylogenetic tree was constructed using the NJ algorithm. The phylogenetic tree branches showed that OMTs from the selected plant species could be divided into class I, class II, and class III, which contained 53, 44, and 23 OMTs, respectively (Fig. 2). Class I contained nine CmOMTs, six CsOMTs, ten SlOMTs, 24 OsOMTs, and four ClOMTs, and it was not included among the AtOMTs (Fig. 2). Class II contained six CmOMTs had 15 AtOMTs, six CsOMTs, four SlOMT genes, nine OsOMTs, and four ClOMTs (Fig. 2). Class III did not contain OsOMTs and was composed of six CmOMTs, two AtOMTs, five CsOMTs, two SlOMTs, and eight ClOMTs (Fig. 2). Multiple species form gene clusters at the base of the NJ tree, with melon, cucumber, tomato, and watermelon showing a closer homology of OMT proteins than Arabidopsis and rice, which may reflect the functional diversification of the OMT gene family after the evolution of dicotyledonous and monocotyledonous plants. In addition, cucurbit OMT proteins (CmOMTs, CsOMTs, and ClOMTs) showed a close homology, suggesting that the OMT family experienced species-specific amplification during the evolutionary process.

Figure 2 Phylogenetic analysis of OMTs from Arabidopsis thaliana, Cucumis melo L., Cucumis sativus L., Solanum lycopersicum L., Oryza sativa L., and Citrullus lanatus (Thumb.).

The structure of a protein determines its function. Therefore, the 3-dimensional structures of CmOMTs were predicted using the PHYRE2 tool using the standard mode. Different percentages of alpha-helices, beta-sheets, and TM-helices were found in the CmOMTs (Fig. S2). Nine CmOMTs (CmOMT7, CmOMT8, CmOMT10, CmOMT12–CmOMT15, CmOMT19, and CmOMT21) had the highest proportions of alpha helices (Fig. S2, Table S4). CmOMT2, CmOMT5 and CmOMT6 contained the highest number of beta-sheets, only CmOMT8, CmOMT9, CmOMT11, CmOMT14, CmOMT16, CmOMT18, and CmOMT21 contained TM-helixes (Fig. S2, Table S4). In addition, disordered regions (DRs) were found in all CmOMTs, and these regions warrant further exploration (Fig. S2, Table S4).

Synteny and GO enrichment analyses of CmOMT genes

Synteny analysis of CmOMTs was performed using the MCScan tool, and the fragment tandem duplication events of the CmOMT gene family members were investigated. As shown in Fig. S3A, only one CmOMT was involved in fragment tandem duplication events among the CmOMT gene family (Table S5). These results suggest that few CmOMTs arose from fragment tandem duplication events; thus, fragment tandem duplication events have a limited evolutionary role in CmOMTs. To further explore the phylogenetic mechanisms of CmOMTs, we constructed syntenic maps of melon based on three species, including a dicot (Arabidopsis thaliana) and two monocots (Citrullus lanatus and Cucumis sativus) (Fig. S3). Multispecies syntenic analysis showed that 2, 6, and 5 CmOMT genes had syntenic relationships with Arabidopsis, watermelon, and cucumber, respectively (Fig. S3). Collinear CmOMT9 and CmOMT14 gene pairs were found between melon and Arabidopsis (Fig. S3B, Table S5). Six collinear CmOMT gene pairs (CmOMT5/9/13/14/20/21) were identified between melons and watermelons (Fig. S3C, Table S5), five collinear CmOMT gene pairs (CmOMT5/9/14/18/21) were associated with cucumber (Fig. S3C, Table S5). Collinear CmOMT9 and CmOMT14 gene pairs were identified between melon and Arabidopsis/watermelon/cucumber, indicating that homologous gene pairs may have been generated before the divergence of monocotyledonous and dicotyledonous plants. Collinear CmOMT5 and CmOMT18 gene pairs were found between melon and other Cucurbitaceae species, suggesting that these OMT genes may have played key roles in the evolution of Cucurbitaceae OMTs.

To obtain a comprehensive understanding of the gene functions, GO term analyses were performed for all CmOMTs. The results of the GO enrichment analysis (Q-value ≤ 0.05) showed the genes corresponded to two biological process classes, namely, ‘methylation’ (Q-value = 6.92e−32) and ‘metabolic process’ (Q-value = 0.0055), and 11 molecular function classes (Table S6), which were closely associated with methyltransferase, such as ‘O-methyltransferase activity’ (Q-value = 2.6e−58), ‘methyltransferase activity’ (Q-value = 1.25e−33), and ‘caffeate O-methyltransferase activity’ (Q-value = 0.0016) (Table S6). These findings were consistent with previous reports showing that the OMT gene family is associated with methylation processes in plants (Liu et al., 2015).

Cis-acting element analysis of CmOMT genes in promoter regions

Cis-acting elements are transcription factor-specific binding sites that play important roles in regulating plant growth, differentiation, and development. We extracted a 2,000 bp sequence in the upstream region of each CmOMT and used the PlantCARE tool to identify the cis-acting elements in CmOMT promoters. Based on functional labeling, 10 cis-acting elements were obtained, among which 50% (5/10) was associated with hormonal responses and 50% was associated with stress and developmental responses (Fig. 3 and Table S7). Thirteen, ten, and seven CmOMT promoters contained ABA-responsive elements (ABREs), auxin-responsive elements (AuxRR-core and TGA-element), and GA-responsive elements (P-box, TATC-box, and GARE-motif), respectively (Fig. 3 and Table S7). MeJA-responsive elements (CGTCA-motif and TGACG-motif) and SA-responsive elements (TCA-element) were present in nine and eight CmOMTs promoters, respectively (Fig. 3 and Table S7). Wound-responsive elements (WUN-motifs) were present only in CmOMT2 and CmOMT8 promoters (Fig. 3 and Table S7). A drought-responsive element (MBS), defense and stress responsiveness element (TC-rich repeats), and low-temperature responsiveness (LTR) element were observed in 7, 8, and 4 CmOMT promoters (Fig. 3 and Table S7). The light-responsiveness element was commonly found in the promoters of all CmOMTs.

Figure 3 Description of cis-elements in CmOMT promoters. Different colors represent different cis-acting elements. Ten cis-acting elements (ABA, auxin, GA, MeJA, SA, wound, drought, light, defense, stress and low-temperature respon.

Expression analysis of CmOMT genes in different tissues and developmental stages

Our results revealed that a large number of development- and stress-related cis-acting elements were widely distributed in CmOMTs. We further explored the expression profiles of CmOMTs in different tissues and developmental stages. Gene expression profiling data were used to analyze the expression of CmOMTs in different tissues and at different developmental stages, and a gene expression heatmap was presented by scaling the expression of each gene across samples (Fig. 4), including the callus, dry seeds, root, stem (stem downside and upside), shoot apex, leaves (young, 6th, 9th and 12th leaves), tendril, flower (anther in male, petal in female, and stigma in female flowers), ovary (DAF 0, 2 and 4), fruit flesh (DAF 8, 15, 22, 29, 36, 43 and 50), and fruit epicarp (DAF 8, 15, 22, 29, 36, 43 and 50) (Table S8). Three CmOMTs (CmOMT16/18/21) were highly expressed in the callus, and only CmOMT14 was expressed in dry seeds. Nine CmOMTs were highly expressed in the roots, e.g., CmOMT5/18/21. Seven CmOMTs were highly expressed in the stems, with CmOMT1/10/18 showing high expression in both the downside and upside of stems. CmOMT9/10/14/18 was highly expressed in leaves (young, 6th, 9th, and 12th leaves), tendrils, and flowers (anther in male, petal in female and stigma in female flower). Notably, CmOMT9 was expressed at higher levels in the petals of female flowers than the other CmOMTs (Fig. 4 and Table S8). CmOMTs that play a key role in fruit pigment methylation can affect plant coloration and improve plant self-protection against environmental stress (Roldan et al., 2014; Du et al., 2015). In the ovary (DAF 0, 2, and 4), CmOMT14/18 was highly expressed. CmOMT1/18 showed an increasing and then decreasing trend during fruit flesh ripening (DAF 8, 15, 22, 29, 36, 43, and 50), consistent with the fruit epicarp. CmOMT14/15 showed an overall increasing trend in fruit flesh (DAF 8, 15, 22, 29, 36, 43, and 50), CmOMT14 displayed an increasing and then decreasing trend during fruit flesh ripening (Fig. 4 and Table S8). Overall, CmOMT1/14/15/18 were highly expressed throughout the growth and developmental stages of melon and in most tissues.

Figure 4 Gene expression heatmap was represented by scaling expression of CmOMT gene across samples under normal growth conditions.

Published RNA-seq datasets were used to evaluate CmOMTs (Yano et al., 2020).

Expression of CmOMT genes in response to abiotic and biotic stresses

To investigate whether CmOMTs are involved in abiotic and biotic stress responses, we analyzed the expression patterns of 21 CmOMTs in varieties with different stress tolerances based on publicly available transcriptomic and qRT-PCR data. The stressors included salt (300 mM NaCl), HTH, cold (6 °C), Fusarium oxysporum f. sp. melonis race 1.2 (FOM1.2), and powdery mildew (Wang et al., 2016; Sebastiani et al., 2017; Zhu et al., 2018; Diao et al., 2020; Weng et al., 2022). The results showed that the expression levels of CmOMT6/9/12/14/18/21 were more variable after salt stress, which could be expanded on in a future study (Fig. 5A). The qRT-PCR results showed that CmOMT6 and CmOMT9 were initially downregulated and then upregulated within 7 d of salt stress, CmOMT12 was significantly downregulated, and CmOMT14/18/21 showed a peak in expression at 3 d after salt stress (Fig. 5D). In summary, CmOMT6/9/14/18 may have a positive regulatory role in response to salt stress, CmOMT14/18 may play a pivotal role in the early stages of salt stress, and CmOMT12 may have a negative regulatory role in response to salt stress. Based on transcriptome data, the results revealed that CmOMT14/18 was highly expressed after HTH stress (Fig. 5B). RT-qPCR analysis indicated that the expression of CmOMT14/18 was induced by HTH stress and that CmOMT14/18 were consistently upregulated after HTH stress (Fig. 5E). Of the 21 CmOMTs that responded to cold stress, four CmOMTs (CmOMT2/12/14/18) were highly expressed under both the control and cold stress conditions, suggesting that these genes may play a potential role in the mitigation of cold stress (Fig. 5D). RT-qPCR analysis revealed that the expression pattern of CmOMT2 was significantly decreased after cold stress and that the expression pattern of CmOMT14/18 was significantly upregulated after cold stress, indicating that these genes may have a positive regulatory role after cold stress (Fig. 5F). Overall, the transcript levels of CmOMT14/18 were significantly affected by multiple abiotic stressors, suggesting a positive regulatory role.

Figure 5 Expression profiles of CmOMT gene family under multiple abiotic stresses.

Volcano plot of 21 CmOMTs in response to salt (A) and HTH (B) stresses based on public RNA-seq data (Wang et al., 2016; Weng et al., 2022). Heatmap of 21 CmOMTs in response to chilling stress (C) based on public RNA-seq data (Diao et al., 2020). qRT-PCR validation of differential expression of CmOMTs involved in salt (D), HTH (E), and chilling (F) stresses. Asterisks and different letters indicate statistically significant differences (P < 0.05).

To explore the potential function of the 21 CmOMTs in response to biotic stress, published RNA-seq data were used to analyze the expression pattern of the CmOMT gene family under FOM1.2 and powdery mildew infection (Sebastiani et al., 2017; Zhu et al., 2018). Fifteen CmOMTs (CmOMT2/3/4/5/7/8/10/11/12/13/14/15/16/18/21) were significantly changed in ‘CHT’ (susceptible cultivar) after FOM1.2 infection, among which seven CmOMTs (CmOMT3/4/5/7/8/16/21) were significantly upregulated at 24 hpi (hours post FOM1.2 inoculation) and five (CmOMT3/5/14/16/18) were significantly upregulated at 48 hpi (Fig. 6A). Similarly, 15 CmOMTs (CmOMT2/3/4/5/6/7/8/9/10/12/13/15/16/18/20) were significantly up/downregulated in the resistant cultivar ‘NAD’ after FOM1.2 inoculation, only one (CmOMT18) was upregulated at 24 hpi, and eight (CmOMT3/4/5/7/8/13/16/20) were upregulated at 48 hpi (Fig. 6A). Interestingly, CmOMT18 reacted to FOM1.2 at an earlier stage in ‘NAD’ than ‘CHT,’ which showed changes at 48 hpi (Fig. 6A). Therefore, CmOMT18 may have a greater potential for FOM1.2 resistance. In addition, to verify whether the CmOMT gene family was involved in the response to FOM1.2, a correlation analysis was performed between resistance genes (PR-1 and LRR genes) and CmOMT s. PR-1 was positively correlated with CmOMT18, and most LRR genes showed a favorable positive correlation with CmOMT18 (Fig. S4). Taken together, CmOMT18 may play a positive regulatory role in the response to FOM1.2. To examine the reaction of the CmOMT gene family to powdery mildew, six CmOMTs (CmOMT3/4/5/7/11/18) were upregulated in MR-1 (resistant cultivar) after powdery mildew inoculation, with CmOMT3/7 upregulated at 1 and 3 dpi (days post-powdery mildew inoculation) and CmOMT5/18 upregulated at 3 and 7 dpi. Most CmOMTs were up/downregulated in the susceptible cultivar ’Top mark’ after powdery mildew inoculation: CmOMT2/9/10/12/15/16 were upregulated at 1 dpi, CmOMT3/9 was upregulated at 3 dpi, and CmOMT3 was upregulated at 7 dpi (Fig. 6B). As CmOMT18 was specifically expressed in leaves, a correlation analysis of the CmOMT gene family with R genes revealed that CmOMT18 may be involved in the regulation of melon resistance to powdery mildew (Fig. S4).

Figure 6 Expression pattern of the CmOMT gene family in response to FOM1.2 (A) and powdery mildew (B).

Asterisks (*) indicate statistically significant differences (P < 0.05).

Discussion

Plants are pedicellates and therefore cannot escape from unfavorable environmental conditions that can affect their growth and development during their life cycle. Unfavorable stress conditions, such as abiotic and biotic stresses, are considered major environmental stressors capable of limiting plant growth and development and directly affecting agricultural yields (Batista-Silva et al., 2019; Yuan et al., 2021). Previous studies have found that secondary metabolites are involved in a variety of biological processes (plant growth, development, and environmental response) and that OMT can catalyze the O-methylation of a variety of secondary metabolites (Liu et al., 2015). OMT genes are widely present in plants such as Arabidopsis, rice, tomato, and watermelon (Lu et al., 2019; Chang et al., 2021; Liang et al., 2022). In this study, we performed a genome-wide prediction of CmOMTs based on the simultaneous detection of conserved OMT structural domains using HMMER, CDD, and SMART tools. We identified 21 CmOMTs, which were named CmOMT1–CmOMT21, in Arabidopsis (17), watermelon (16), and Citrus (58) (Lu et al., 2019; Chen et al., 2021; Chang et al., 2021). Genomic DNA, CDS length, and deduced amino acid sequences varied among the CmOMT genes, leading to variations in the theoretical molecular weight, PI, instability index, aliphatic index, and GRAVY of CmOMTs (Table 1). CmOMTs were unevenly distributed on all chromosomes, with most CmOMTs on chromosome 1 (Fig. 1 and Table 1). Similar CmOMT distribution patterns have been observed in other plants, such as Arabidopsis, watermelon, Citrus, and rice (Lu et al., 2019; Chen et al., 2021; Chang et al., 2021; Liang et al., 2022).

The phylogenetic relationship analysis showed that OMT genes could be clustered within a monophyletic group derived from non-plant genes. Based on functional traits, the results indicated the existence of two plant serial group clusters (Lam et al., 2007). In this study, 120 OMT proteins were divided into three branches. Although different from the evolutionary analysis results of other species, the OMT genes of cucurbit crops, such as watermelon, were similarly divided into three categories, which may be due to tandem duplication (Lu et al., 2019; Chang et al., 2021; Chen et al., 2021; Liang et al., 2022). The presence of ancient duplication events and high retention rates has led to the existence of a large number of duplicated genes in plant genomes. These duplicated genes have contributed to the evolution of genes with novel functions, such as the production of floral structures, enhanced disease resistance, and adaptation to different adversities (Nicholas, Melissa & Shin-Han, 2016; Zhang et al., 2020). Cucurbitaceae OMT proteins may have undergone more favorable changes during evolution, thereby generating new functions (Chang et al., 2021). Of the three branches, class I had the most OMT members, with 53 OMT proteins, although Arabidopsis had nine CmOMTs, four ClOMTs, ten SlOMTs, six CsOMTs, and 24 OsOMTs. These findings were similar to the results of a study on watermelon (Chang et al., 2021). Differences in the phylogenetic relationships between plant OMT homologous proteins may reflect the diversity of their functions in growth, development, and environmental responses. In Arabidopsis, the OMT protein AtOMT3 is a class III protein that has close homology with CmOMT9. AtOMT3 is defined as AtASMT1, which increases melatonin levels, improves plant salt tolerance, and is closely related to Pst DC3000 resistance (Shi, Wei & He, 2016; Zheng et al., 2017). AtOMT6 is an OMT protein on class II OMT protein that shows close homology with CmOMT8 and has been identified as AtCOMT1, which is associated with melatonin synthesis, regulating melatonin synthesis levels, and improving resistance to multiple stresses in plants, such as salt stress and heat stress (Zheng et al., 2017; Hasan et al., 2023). In contrast, Chang et al. (2021) reported that ClCOMT1 (ClOMT3) is also closely homologous to CmOMT18 and could enhance abiotic stress resistance in Arabidopsis by increasing endogenous melatonin content. For class I OMT proteins, Liang et al. (2022) reported that OsCOMT8, OsCOMT9, and OsCOMT15 play key roles in lignin synthesis. Furthermore, we observed that CmOMTs presented a closer relationship with the OMTs of watermelon, cucumber, and tomato than those of Arabidopsis and rice in each branch. This finding implies that the OMT gene family of dicotyledons and monocotyledons experienced functional diversification during their long-term evolution. This hypothesis is partially supported by the observation that OMT transcript abundance and expression patterns in Arabidopsis and rice differ significantly from those in watermelon and melon under normal growth conditions (Lu et al., 2019; Chen et al., 2021; Chang et al., 2021; Liang et al., 2022).

One of the relatively reliable parameters for evaluating the evolution of gene families is variation in structure or motifs. Analyses of plant evolutionary relationships have shown that genes with similar intron-exon structures and conserved motif arrangements often have similar functions (Long, Souza & Gilbert, 1995). Most members of the same taxon of the 67 OMTs in soybean have similar gene structures, and the same results have been observed in pomegranate and rice (Zhang et al., 2021a; Zhang et al., 2021b; Zhao et al., 2022; Liang et al., 2022). In our study, 21 CmOMTs were classified as class I, class II, and class III, which consisted of nine, six, and six members, respectively. This classification is consistent with observations in watermelons (Chang et al., 2021) and is further supported by the fact that cucurbits are different from other species. Analysis of the differences in exon-intron patterns of CmOMTs among the three classes showed that most of class I contained 1–2 exons, class II contained 2–3 exons, class III contained more than three exons, and the intron region was longer in class III. Similar exon-intron patterns within the same phylogenetic class may be associated with the tandem duplication of these sequences (Du et al., 2015).

GO enrichment and cis-acting element analyses are important methods for gene function studies and have been widely used to predict the possible biological functions and regulatory patterns of gene sets of interest in organisms (Wang et al., 2022). We performed GO enrichment analysis of the CmOMTs and identified a total of 13 overrepresented terms (Q-value ≤ 0.05), and they were mainly related to methylation in the biological process category and methyltransferase activity in the molecular function category. The results were similar to the functions of OMTs in soybean and watermelon, thus demonstrating the similar functions of OMTs in different species (Chang et al., 2021; Zhao et al., 2022). The specific binding of cis-acting elements with transcription factors in the promoter to regulate gene expression is not only the most important method of biological signal transduction but also an important means by which genes interact with other genes (Cheng et al., 2019). To better understand the role of CmOMTs in the response to abiotic and biotic stresses, we analyzed the type, number, and distribution of cis-acting elements of 21 CmOMTs in the promoter. Light-responsive elements were present in all CmOMTs, suggesting that the expression of CmOMTs may be regulated by light, similar to previous studies on soybeans (Zhao et al., 2022). In addition, the number of hormone-response cis-acting elements and stress-response cis-acting elements suggests that CmOMT family members may be extensively involved in growth, development, and environmental stress. OMTs have been associated with enhanced salt stress resistance in Arabidopsis, tomatoes, and watermelons (Chun et al., 2019; Liu et al., 2019; Chang et al., 2021). Moreover, they are reactive to drought and low-temperature stress (Sun et al., 2019; Chang et al., 2021) and involved in disease resistance, including Pst DC3000 and Xoc (Zhao et al., 2015; Ahammed et al., 2020).

Gene duplication has occurred in 70–80% of angiosperms and represents a method of generating new genes and responding to environmental stress (Panchy, Lehti-Shiu & Shiu, 2016; Qiao et al., 2019). Five pairs of duplicated genes out of the 21 CmOMTs suggested that tandem duplication events greatly contributed to the expansion of the CmOMT gene family and a similar result was also obtained for tomato (Sun et al., 2021). To examine the phylogenetic relationships of OMT genes between melon and other plants, collinear relationships between melon and Arabidopsis, watermelon, and cucumber were investigated. The number of collinear events between melon and watermelon and cucumber was much greater than that between melon and Arabidopsis, which is consistent with the smaller evolutionary distance between genera of the same family (Sun et al., 2021).

Analyses of gene expression profiles in plants can reveal the functions of genes. For example, ClOMT7 is expressed only in the root while OsCOMT7 is expressed at the highest level in the stem (Liang et al., 2022; Chang et al., 2021). Moreover, the expression patterns of OMT genes vary among different species. Therefore, our tissue- and development-specific expression profiles of CmOMTs provide a basis for determining the involvement of CmOMTs in plant physiological and biochemical activities. Previous studies showed that OMT genes play important roles in abiotic and biotic stress responses and are key enzymes for the synthesis of lignin and the regulation of plant physiological responses to stresses, such as drought, salt, and high or low temperatures (Hamada et al., 2004) They also play a role in plant resistance to pathogenic invasion by regulating the synthesis of secondary metabolites (Maury, Geoffroy & Legrand, 1999; Zhao et al., 2004; Kim et al., 2007). Analysis of transcriptome and RT-qPCR data under salt, low temperature, and HTH stresses showed that seven CmOMTs were involved in the stress response to different degrees. The CmOMT gene family plays an important role in some abiotic stress responses, among which the transcript levels of CmOMT14 and CmOMT18 showed an increasing trend under all three stress conditions, indicating that they play important roles under abiotic stress conditions. Fifteen CmOMTs were significantly altered under FOM1.2 infection, and 12 CmOMTs were significantly upregulated under powdery mildew infection in both resistant and susceptible melon varieties. These results suggest a role for CmOMTs in the response to certain biotic stresses. The correlation of CmOMT genes with R genes showed that CmOMT18 was more correlated with R genes under FOM1.2 and powdery mildew infection. The OMT phylogeny analysis among plant species also indicated that CmOMT18 is closely homologous to AtOMT16 (AtCOMT1) and ClOMT3 (ClCOMT1) and CmOMT18 may be related to AtOMT6 (AtCOMT1) and ClOMT3 (ClCOMT1) based on the homology among proteins under biotic and abiotic stress (Zheng et al., 2017; Chang et al., 2021).

Conclusions

In conclusion, OMT gene families of melon and five other representative plants were investigated in this study to identify their roles in biotic and abiotic stress. We identified the chromosomal locations, phylogenetic relationships, gene structures, and conserved motifs of CmOMTs and determined the expression patterns of CmOMTs in response to abiotic and biotic stresses via RNA-seq and RT-qPCR. These results not only provide new insights into the characteristics of the plant OMT family but will also facilitate functional genomic investigations of CmOMT genes in future studies.

Supplemental Information

Supplemental Information 1 List of quantitative real-time PCR (qRT-PCR) primers for expression analysis of CmOMT genes

Click here for additional data file.

Supplemental Information 2 Identification of conserved domains in CmOMT proteins using the HEMMER tool

Click here for additional data file.

Supplemental Information 3 Information about OMT proteins in Cucumis sativus L

Click here for additional data file.

Supplemental Information 4 3-D structures of CmOMT proteins predicted using the PHYRE2 tool

Click here for additional data file.

Supplemental Information 5 Orthologous relationships between melon and Arabidopsis/watermelon/cucmber

Click here for additional data file.

Supplemental Information 6 List of GO (gene ontology) terms of CmOMT genes

Click here for additional data file.

Supplemental Information 7 Identification of cis-regulatory elements in the promoters of CmOMT genes

Click here for additional data file.

Supplemental Information 8 Tissue space expression profile of CmOMT genes under normal growth conditions

Published RNA-seq datasets were used to evaluate CmOMT genes (Yano et al., 2020).

Click here for additional data file.

Supplemental Information 9 Bit score represents the information content of five conserved domains mapped using the MEME tool (A)

Position and multiple sequence comparisons of the five conserved structural domains among CmOMTs, with different colors indicating

Click here for additional data file.

Supplemental Information 10 Predicted 3-D structures of CmOMT proteins

Click here for additional data file.

Supplemental Information 11 Schematic diagram of the synthesis of CmOMTs

(A) Interchromosomal relationships between CmOMTs in melons. Synteny analysis of OMTs between melon and Arabidopsis (B), watermelon (C), and cucumber (D). Gray lines indicate collinear gene pairs and red lines represent collinear OMT gene pairs. C. melo, A. thaliana, C. lanatus, and C. sativus represent melon, Arabidopsis, watermelon, and cucumbers, respectively.

Click here for additional data file.

Supplemental Information 12 Correlation analysis between CmOMTs and resistance genes

Click here for additional data file.

Supplemental Information 13 Raw data of qPCR

Click here for additional data file.

Additional Information and Declarations

Competing Interests

Author Contributions

Data Availability

The authors declare there are no competing interests.

Shuoshuo Wang conceived and designed the experiments, performed the experiments, analyzed the data, prepared figures and/or tables, and approved the final draft.

Chuang Wang analyzed the data, prepared figures and/or tables, and approved the final draft.

Futang Lv analyzed the data, prepared figures and/or tables, and approved the final draft.

Pengfei Chu analyzed the data, prepared figures and/or tables, and approved the final draft.

Han Jin conceived and designed the experiments, analyzed the data, authored or reviewed drafts of the article, and approved the final draft.

The following information was supplied regarding data availability:

The raw measurements are available in the Supplementary File.

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
