# Peer review of "Genome-wide identification of the OMT gene family in Cucumis melo L. and expression analysis under abiotic and biotic stress"

_PeerJ, doi:10.7717/peerj.16483_

## Round 0.1 · original submission · Major Revisions

The authors need to include all suggestions given by reviewers.

Reviewer 1 ·

Basic reporting

In the manuscript entitled “Genome-wide identification of OMT gene family in Cucumis melo L. and their expression analysis that respond to abiotic and biotic stresses”, authors provided a biological information basis of 21 CmOMTs by phylogenetic analysis, gene structure analysis, cis-regulatory elements analysis and so on. In addition, the expression pattern of 21 CmOMTs under abiotic and biotic stresses were studied using the published RNA-seq dataset. The manuscript provided a theoretical and practical basis for further studying the molecular mechanism of CmOMTs in melon.

Experimental design

1. Please state the basis for selecting PF00891 as the reference sequence, or add references in line 129.
2. For expression profiles of Fig.6 and 7, the expression level of each gene should be compared in different tissues or treatments rather than standardizing the expression data of all genes and all tissues. For example, it is difficult to distinguish the expression difference of CmOMT12 or CmOMT19 in different tissues except for DAF0 in the current version of Fig.6.
3. To distinguish the positive and negative correlations, the color of negative correlations can be changed into blue in Fig. S2.

Validity of the findings

1. The protein sequences of OMT homologs were applied for phylogenetic analysis, but the author repeatedly stated the phylogenetic relationships of OMT genes. Please modify this description.
2. In line 203: 2 CmOMT genes on chromosome 4 and 11, only one CmOMT gene on chromosome 7 and 11 (Fig. 1A). The conclusion for chromosome 11 was wrong.
3. In line 252: Using the MEME online tool, protein conserved motifs were identified in CmOMT genes. “CmOMT genes” were mischaracterized.

Additional comments

1. The species name of FAOMT and VvAOMT should be stated in line 63.
2. Genes should be italicized, otherwise proteins. Multiple errors existed for the format of CmOMT such as line 21, 147. In addition, ‘cis’ also should be italicized. There was a spelling mistakes in line 324 “patters”. The manuscript should be thoroughly reviewed.
3. The legend of Fig.3C was missing.

·

Basic reporting

In this study, the authors have conducted integrative studies on 21 O-Methyltransferases (OMTs) from melon. These studies encompass protein, gene, and promoter sequence analyses. Additionally, they have investigated the gene expression of OMTs under various abiotic and biotic stress conditions. While this research has the potential to be a valuable contribution, it currently contains several technical and fundamental issues that would have benefitted from proofreading by a fluent English speaker before submission. However, despite these important aspects, they should not deter the authors from revising their work.

The manuscript requires a comprehensive revision by a fluent English speaker as it contains numerous sentences that are awkwardly constructed, ranging from the abstract to the discussion section. Although there are too many instances to point out individually, one example is the following sentence from the abstract: "O-Methyltransferases (OMTs) have been thought to methylation capabilities, which could increase the diversity and stability of secondary metabolites, and improve plant protection from environmental stresses." This sentence is not only vague but also fails to convey comprehensible information. Similarly, a sentence from the discussion section states, "In course of long-term evolution, tandem replication genes were usually determined by their function. Tandem replication genes were formed owing to differential subfunctionalization and de novo functionalization, and play an important role in plant growth, development, and environmental responses (Zhang., 2003)."

The results should be tested against the background or random chance whenever necessary. The authors should use standard gene names for melon, as well as genes from other species, unless they confirm the evolutionary relationship of each gene with known OMT genes with statistical significance and comprehensive homology analysis. Even then, there should be a rationale provided for using different names, as they could interfere with computational algorithms that extract automatic information from the literature.

Here are some important points to improve the manuscript.

Figure 1 displays the chromosomal locations and amino acid composition of the 21 OMTs. I have a question: what is the inference drawn from the amino acid composition? It would be more informative to have the first figure depict the protein domain architecture and the gene structure of the 21 OMTs, and it should be first attempt to classify them into groups. See PMID: 30698789 for example.

Figure 2 presents a phylogenetic tree, but the bootstrap values are given as fractions, which is unusual. I would prefer to see both gene and protein sequence-based phylogenetic trees side by side, with the original gene names included. It also makes sense to classify the 21 genes into subclasses first before studying them alongside sequences from other species. Therefore, it would be better to reorganize Figures 1, 2, and 3 accordingly.

In Figure 3, please show the UTR, exon, and intron instead of just UTR and CDS. The phylogenetic tree shown in Figure 2 can also be presented in this manner. I think, it would be better if authors use intree, RAxML or Mr. Bayes like tree inference software which actually construct phylogenetic tree what users are showing is a clustering dendrogram.

Figure 4 could be included in the supplementary material, as its conveyed message is unclear to me.

Regarding Figure 5, I only observe a random distribution of identified cis-regulatory elements in the gene promoters. It is not convincing if transcription factor binding sites (TFBS) are scattered randomly throughout the promoter region. This occurrence could be purely coincidental. TFBS should be strategically positioned relative to the TSS, and this is not apparent here. The authors should demonstrate which of these sites cannot be predicted by random chance alone and should also show that these transcription factors exhibit expression correlation with the respective genes.

For Figures 6, 7, and 8, heatmaps or barplots lack meaning if there is no statistical significance in expression differences. What I would like to see for each gene is boxplots depicting expression in control and treated conditions, accompanied by p-values indicating statistical differences.

Experimental design

It is crucial to conduct phylogenetic and gene expression analyses using appropriate statistical significance tests. Without these tests, it becomes challenging to distinguish the results from random chance. Incorporating statistical significance tests will provide the necessary framework for evaluating the reliability and significance of the obtained findings.

Validity of the findings

Although the authors attempted to validate their findings with RT-PCR, I struggled to comprehend the information due to the poor language and presentation of the results. It is important for the authors to improve the clarity and organization of their findings to ensure that the results can be effectively understood and interpreted. Additionally, they should consider revising the language used to describe the experimental procedures and results to enhance readability and comprehension.

·

Basic reporting

This study describes a genome-wide identification of OMT gene family members in the melon genome database in comparison with other species. 21 family members were identified and their phylogenetic relationship, location in chromosome, gene structure and amino acid composition were studied. Additionally, expression analysis was performed to determine tissue-specific expression and response to abiotic and biotic stresses.
These findings could be useful to identify and study stress-related genes.
The work is in general well performed and provides useful insight on the subject matter.
The English language, however, should be improved considerably throughout the text to ensure that readers can clearly understand the contents and contributions of the article. A thorough review of the article by a person who is proficient in English or by a professional editing service is necessary to increase the overall quality of the manuscript.

Experimental design

No comment

Validity of the findings

No comment

Additional comments

Minor points.
Abstract is longer than necessary and contains non-essential information

Introduction: Presents an ample introduction of the gene family, the functions of the enzymes etc. but it does not provide information about the evolutionary history of the genes or the gene family in related species, which is pertinent to the present study.

Specific comments:
Line 116 28ºC
Line 119 Indicate how HTH treatment differs from standard culture
Line 197-198 It is not necessary to indicate here the tools and methods used and e value.
Same comment is valid throughout Results section

Line 197 Paragraph that begins at this line is not clear. This is important as it sets the whole scheme for the description of the results.

Discussion is longer than necessary

---

## Round 0.2 · Minor Revisions

The authors need to include suggestions given by reviewer 2 before re-submission of revised version.

Reviewer 1 ·

Basic reporting

This manuscript has been thoroughly reviewed and corrections have been incorporated.

Experimental design

Sufficient.

Validity of the findings

Robust.

·

Basic reporting

Manuscript reads much better after revision. There are still some minor issues as listed below and should be resolved before acceptance of the manuscript.

1) Authors should use original gene names or locus names along whenever they use their own gene nomenclature unless they show evidence that those gene names are universally accepted or used in one of the major public databases.

2) Gene expression heatmap in figure 4 may be represented by scaling expression of each gene across samples

3) It is not clear how statistical significance was calculated for the gene expression between groups.

Experimental design

Figure 5 A, B, C heatmaps should be replaced with volcano plots, where one can see the p-value and log-fold change between groups. Please use limma or t-test or wilcox-test to compute significance

Figure 5 D, E, F, ; Samples from the Day 0 and the hour 0 should be used as a reference to compute significance with remaining groups. Instead of letters, represent asterisks to show significance, higher the significance more the number of asterisks.

Figure 6. It is not clear what those asterisk marks actually represents. Please write what statistical test was used to evaluate expression difference.

Validity of the findings

Findings were validated using experiments.

Additional comments

There is still issues with terms through out manuscript.

For example

line 186, Replication word is not appropriate with tandem, it is much better to use tandem duplication throughout the manuscript

line 201, "The phylogenetic tree was optimized using Figtree v1.4.3", optimzed should be replaced with visualized.

line 326, please replace "simulated" word with "predicted"

---

## Round 0.3 · accepted · Accept

The revised manuscript is good to accept for publication.

Reviewer 1 ·

Basic reporting

This manuscript has been thoroughly reviewed and corrections have been incorporated. No additional comments.

Experimental design

No additional comments.

Validity of the findings

No additional comments.

·

Basic reporting

The authors have adequately addressed all the concerns in the revised manuscript, and I no longer have any reservations.

Experimental design

no comment

Validity of the findings

no comment

Additional comments

no comment

·

Basic reporting

No comment

Experimental design

No comment

Validity of the findings

No comment

Additional comments

The new version of the manuscript is greatly improved.
It addresses all my previous concerns.